# EEG Beta functional connectivity decrease in the left amygdala correlates with the affective pain in fibromyalgia: A pilot study

**Soline Makowka**[1], **Lliure-Naima Mory**[1,2], **Michael Mouthon**[1], **Christian Mancini**[1], **Adrian G. Guggisberg**[3], **Joelle Nsimire Chabwine**[1,2]*

**1** Faculty of Science and Medicine, Department of Neuroscience and Movement Science, Laboratory for Neurorehabilitation Science, Medicine Section, University of Fribourg, Fribourg, Switzerland, **2** Neurorehabilitation Division, Fribourg Hospital Meyriez/Murten, Fribourg, Switzerland, **3** Department of Clinical Neuroscience, Division of Neurorehabilitation, Geneva University Hospital, Geneva, Switzerland

☉ These authors contributed equally to this work.
* Joelle.chabwine@unifr.ch

**Data Availability Statement:** Patients have neither given their consent to share their coded data on a public repository nor to anonymize their data for

## Abstract

Fibromyalgia (FM) is a major chronic pain disease with prominent affective disturbances, and pain-associated changes in neurotransmitters activity and in brain connectivity. However, correlates of affective pain dimension lack. The primary goal of this correlational cross-sectional case-control pilot study was to find electrophysiological correlates of the affective pain component in FM. We examined the resting-state EEG spectral power and imaginary coherence in the beta (β) band (supposedly indexing the GABAergic neurotransmission) in 16 female patients with FM and 11 age-adjusted female controls. FM patients displayed lower functional connectivity in the High β (Hβ, 20–30 Hz) sub-band than controls ($p = 0.039$) in the left basolateral complex of the amygdala ($p = 0.039$) within the left mesio-temporal area, in particular, in correlation with a higher affective pain component level ($r = 0.50$, $p = 0.049$). Patients showed higher Low β (Lβ, 13–20 Hz) relative power than controls in the left prefrontal cortex ($p = 0.001$), correlated with ongoing pain intensity ($r = 0.54$, $p = 0.032$). For the first time, GABA-related connectivity changes correlated with the affective pain component are shown in the amygdala, a region highly involved in the affective regulation of pain. The β power increase in the prefrontal cortex could be compensatory to pain-related GABAergic dysfunction.

## Introduction

Fibromyalgia (FM) is one of the most frequent chronic pain disease, reaching up to 4% of frequency in the general population, and affecting more significantly females than males [1]. The clinical constellation characterizing FM combines widespread chronic pain, fatigue, mood disorders, cognitive deficits and sleep disturbances. Despite active research on underlying mechanisms and etiologies, FM remains a poorly understood disease condition.

As in any other chronic pain syndrome, central sensitization plays an important role in FM [2], which could be reflected by altered brain dynamics in several areas involved in nociception

such purpose, as confirmed by the Ethical Committee of Vaud (CER-VD). Prof Dominique SPRUMONT, the President of the CER-VD (e-mail: dominique.sprumont@vd.ch or secretariat.cer@vd. ch), is ready to answer any query regarding this issue (please mention the study number PB_2016-00739 (331/15) in each related correspondence to the CER-VD).

**Funding:** A minor part of this study was jointly supported by the University of Fribourg, the Fribourg Hospital and the Quadrimed Fund. The funders had no role in study design, data collection and analysis, decision to publish, or preparation of the manuscript.

**Competing interests:** The authors have declared that no competing interests exist.

observed in functional connectivity (FC) studies [3]. A recent resting-state fMRI study found transient functional connectivity changes related to ongoing pain intensity, driven by central sensitization in brain areas responsible for pain regulation in FM patients [4]. However, the study did not discriminate between different dimensions of pain, which could have contributed to further understand mechanisms subtending observed connectivity modifications. Knowing the prominent impact of emotional disturbances in FM, we made the hypothesis that the affective pain component would be related to connectivity changes in emotional pain-regulating brain regions [4].

Chronic pain induces significant modifications in neurotransmitter pathways, the most remarkable among them being an over-excitatory state owing to a decrease in the inhibitory input [5] and/or excessive excitatory neurotransmission [6]. Accordingly, substantial decrease in brain GABAergic signaling is reported in chronic pain patients [7], brain inhibition being mainly driven by GABAergic interneurons [8]. Furthermore, we recently showed reduction in EEG markers of the GABAergic neurotransmission, namely beta (β) oscillations, in chronic neuropathic pain [9]. Thus, we further assumed that expected power and FC changes in FM would be measurable through decrease in the EEG β oscillatory band.

This pilot study is part of a larger project investigating β EEG oscillations considered to be indicators of the GABAergic neurotransmission in chronic pain clinical models as a contribution to the mechanistic approach of pain characterization and therapy. In this particular investigation, the aim was to assess EEG FC changes in the β frequency domain occurring in FM patients and relate observed modifications to the affective component of pain.

## Materials and methods

### Study design

This correlational, cross-sectional case-control pilot study included FM patients and age- and sex-adjusted healthy participants (2018–2020). Approval by the Ethical Committee of Vaud (CER-VD) was obtained under the number PB_2016–00739 (initial number 331/15). Each participant signed an informed consent form prior to any data collection and received a financial compensation thereafter.

Patients were recruited mainly through neurologists, rheumatologists and pain specialists from Fribourg Hospital, and through Swiss FM associations using web-based and flyer advertisements, while advertisements for controls targeted middle-aged adult hobby associations. Participants (cases and controls) were adult (≥18 y) females [1] and right-handed [9]. The diagnosis of FM had to be made by the specialists and meet internationally admitted criteria (see below). Exclusion criteria consisted in: existence of central nervous system lesion or disease (including epilepsy and parasomnia), significant cognitive impairment, coexistence of any other type of pain (patients) or any pain (controls), and surgery involving any nervous system structure less than six months before inclusion [9].

### Data collection

Each participant was interviewed following a standardized questionnaire (age, sex, marital status, profession and education, treatments, relevant medical history) and underwent a brief neurological examination in order to exclude abnormalities potentially related to a central nervous system lesion or disease.

Three additional specific questionnaires (the FM Rapid Screening Tool (FiRST) [10], the Symptoms Severity Score (SSS) and the Widespread Pain Index (WPI) of the 2010 American College of Rheumatology criteria (ACR 2010) were used to confirm FM (FiRST > 5, SSS ≥ 7 and WPI ≥ 5 or SSS ≥ 9 and WPI 3–6 [11]).

Pain evaluation (FM patients) included pain intensity using the Visual Analogue Scale (VAS) on the day of evaluation ($VAS_d$) and on average over the week before [12], considering $VAS_d \geq 3$ as significant [9]. The VAS shows good statistical qualities for evaluation of chronic pain patients [13–15], and is frequently used in FM studies [16,17]. The Short-form 2 McGill Pain Questionnaire allowed differentiation between the sensory ($SF-MPQ-2_{sensory}$) and the affective ($SF-MPQ-2_{affective}$) components of pain (both scores re-scaled /10) [18]. The SF-MPQ-2 has been developed to be used in chronic pain populations and has excellent reliability and validity [18]. It has also been often used in FM studies [17,19–21]. The Hospital Anxiety and Depression Scale (HADS) determined existence of anxiety and depression [22], whereas the Insomnia Severity Index (ISI) was compiled for insomnia assessment [23]. HADS is a reliable instrument for screening clinically significant anxiety and depression as well as their severity [22,24], frequently used in FM studies [25–27]. Finally, ISI is a reliable and valid instrument to quantify insomnia severity [23,28]. It has already been used in chronic pain population [29,30], including in FM patients [31–33].

The EEG data were recorded using a high density 64-channel EEG recording system (BIOSEMI ActiveTwo, Amsterdam, Netherlands) at a sampling rate of 1024 Hz. All EEG recordings were obtained before noon [9] in a quiet dark room shielded by a Faraday cage. Participants were requested to sit down with eyes closed, minimizing eye blinks and body movements. The recording lasted 24 minutes. To avoid sleepiness due to the length of the recording, participants were maintained seated, and acoustic sounds were used two times during the recording.

**EEG data.** Raw EEG data were down-sampled to 512 Hz, and band-pass filtered between 0.5 and 40 Hz. Bad EEG channels were excluded by visual inspection using Cartool software for data visualization, while careful manual artifact-rejection was performed to exclude eye movements and blinks, body movements and electrode drifts. Only the first five minutes of artifact-free data of the recording was retained for the analysis.

Preprocessed data were referenced to the Cz electrode and segmented into non-overlapping 1-second epochs. Analyses were performed in MATLAB (The MathWorks), using the toolbox NUTMEG [34,35]. The lead-potential was computed using a boundary element head model [36,37], with the Helsinki BEM library [34] and the NUTEEG plugin of NUTMEG. The head model was based on the Montreal Neurological Institute template brain, and solution points were defined in the gray matter with 10 mm grid spacing.

EEG epochs were Hanning-windowed, Fourier transformed, and projected to gray matter voxels, using an adaptive filter (scalar minimum variance beamformer) [38] and the δ (0.5 to 3.5 Hz), θ (3.5 to 7.5 Hz), α (7.5 to 12.5 Hz), Low β (Lβ, 13–20 Hz) and High β (Hβ, 20–30 Hz) frequency bands were defined. The β band (13–30 Hz) [39], supposedly indicating brain GABAergic activity, was divided into Lβ and Hβ sub-bands following our previous observations [9,40,41], whereas the δ band was considered as a control frequency.

The absolute source spectral power was computed as the absolute squared signal amplitude, whereas the relative power was obtained by normalizing the power in each band to the mean power of all bands and dividing by their standard deviation; thus obtaining z-scores.

FC was assessed in source space (i.e. after source localization) as the statistical dependency between reconstructed activities at the different solution points. Analysis of FC was conducted as described previously [34,42]. We used the absolute imaginary component of coherence as index of FC and calculated the weighted node degree (WND) for each solution point as the sum of its coherence with all other cortical solution points [43]. In order to minimize EEG signal-to-noise ratio influence on FC, we normalized WND values using z-scores by subtracting the mean WND value of all voxels of the subject from the imaginary component of coherence values at each voxel and by dividing by the standard deviation over all voxels [44,45].

**Statistical analysis.** Statistical non-parametric mapping was used to compare patients to controls at all solution points of the cortex. Correction for multiple testing was obtained by defining a cluster-size threshold based on the cluster size distribution obtained after random reversions of original data [46]. This voxel-wise analysis revealed the topography of contrasts, which was complemented by anatomical region of interest (ROI) defined with the Julich atlas [47], and the later thereafter compared between patients and controls using an unpaired $t$-test.

Associations between EEG and clinical data were analyzed with Spearman correlation test (more robust to detect outliers than Pearson correlation). Data are all presented as mean (SD) and the level of significance admitted at $p<0.05$ (95% confidence interval).

Voxel-wise statistics were performed with the toolbox NUTMEG, the remaining analyses with the Statistics toolbox of MATLAB (The MathWorks) [34,35].

# Results

## General data

In total, 16 patients and 11 controls (51.8(8.5) and 54.2(4.6) years) were included for the analysis, as shown in the selection flowchart (Fig 1). FM patients complained of the typical widespread pain at moderate intensity the day of evaluation ($VAS_d$ 4.75(2.84)) and during the last week before assessment (6.38(2.11)). The SF-MPQ-2$_{affective}$ and SF-MPQ-2$_{sensory}$ scores were similar (5.88(2.87) and 5.29(2.33), respectively). Patients reported moderate insomnia (ISI 17.75(5.27)), anxiety (HADS anxiety sub-score 10.50(4.12)) and depression (HADS depression sub-score 10.94(3.07)), while controls had normal scores (respective $p$ <0.001, <0.005 and <0.001). Information regarding the patient's medications are detailed in Table 1.

## Spectral power analysis

Absolute source power values displayed no difference between FM patients and controls, while a significant cluster of higher Lβ relative power was observed in FM patients in the left

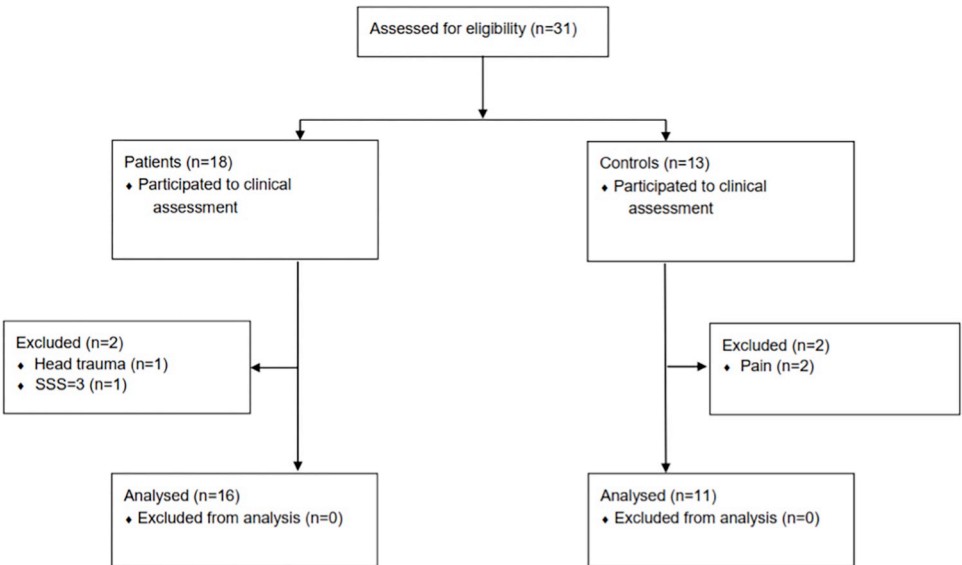

**Fig 1. Participant selection procedure.** In total, 31 participants were screened (18 fibromyalgia patients and 13 controls). Two patients were excluded because one had head traumatism and the other a symptoms severity scale (SSS) of 3 (the lowest SSS limit for fibromyalgia diagnosis was 7). Two controls were excluded as they complained of pain the day they were assessed. Finally, 16 patients and 11 controls were included in the study.

**Table 1. FM patient's medications.**

| TREATMENT | Type of treatment | n/16 |
|---|---|---|
| | NSAID* | 7 |
| | Antidepressants | 7 |
| | Physical and alternative | 5 |
| | Antimigrainous | 4 |
| | Benzodiazepines | 3 |
| | Opiates | 2 |
| | Other drugs | 5 |
| | None | 3 |

*NSAID: Nonsteroidal Anti-Inflammatory Drugs.

prefrontal cortex (PFC) (Fig 2A), exclusively correlated with the $VAS_d$ ($\rho = 0.54$, $p = 0.032$) (Fig 2B). FM patients with $VAS_d \geq 3$ displayed higher relative $L\beta$ power than those with $VAS_d < 3$ ($p = 0.028$). Neither difference nor correlation were seen in the $\delta$ band.

### Functional connectivity

A significant decrease in $H\beta$ FC was noticed in FM patients compared to controls in the left mesiotemporal area (Fig 3A), with a trend to correlation with the SF-MPQ-2$_{affective}$ ($\rho = 0.45$, $p = 0.082$) (Fig 3B). Within the left mesiotemporal area, the basolateral amygdala (BLA) displayed a significant decrease in $H\beta$ FC ($p = 0.039$; Fig 3C and 3D), and a significant correlation with the SF-MPQ-2$_{affective}$ ($\rho = 0.50$, $p = 0.0495$), with no impact of ongoing pain. No FC difference appeared in $L\beta$ and $\delta$ frequencies, and the $H\beta$ FC did not correlate with the other clinical scores.

## Discussion and conclusion

The most remarkable result of this study is the decrease of FM patients' $H\beta$ FC in the BLA and its selective correlation with the affective pain component. For the first time, a clear anatomo-

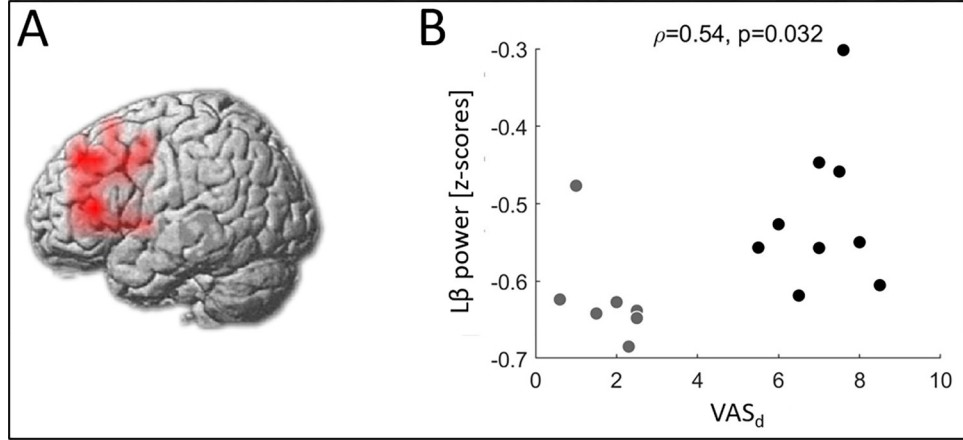

**Fig 2. Low β (Lβ) relative power in fibromyalgia (FM) patients vs. controls.** A voxel-wise analysis of the entire cortex (see methods for details) revealed a significant cluster of increased $L\beta$ band (13–20 Hz) relative power in FM patients compared to controls (respective mean(SD)) of -0.56(0.10) and -0.69(0.06)) in the prefrontal cortex (red color, $p < 0.05$, cluster corrected) **(A)**. This increase correlated with the ongoing pain intensity ($VAS_d$) **(B)**. Grey dots correspond to patients with $VAS_d < 3$ whereas black dots are related to patients with significant pain ($VAS_d \geq 3$).

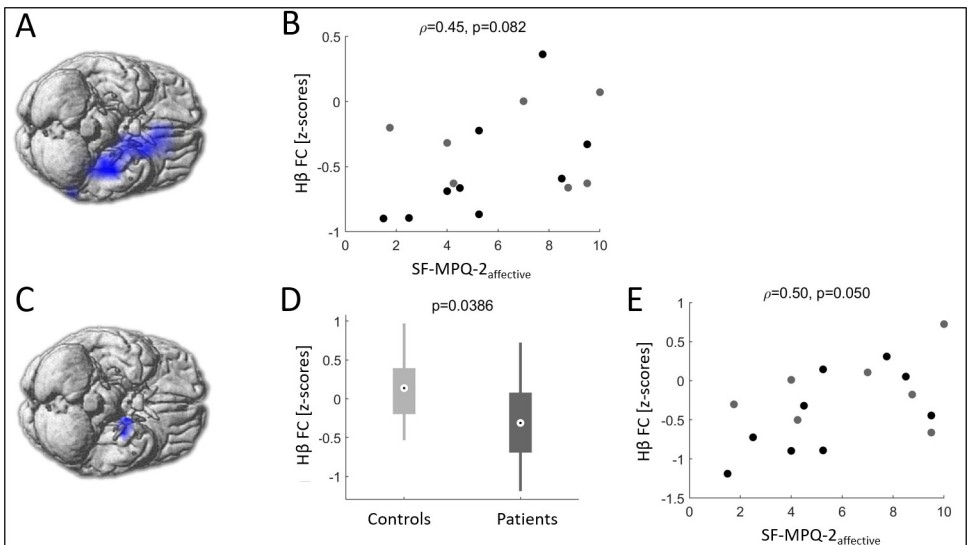

**Fig 3. High β (Hβ) functional connectivity (FC) in fibromyalgia (FM) patients vs. controls.** The voxel-wise analysis (see methods for details) revealed decreased Hβ (20–30 Hz) FC in FM patients in comparison to controls in the left mesiotemporal area (blue color, $p<0.05$, cluster corrected) **(A)**. A correlation was observed between Hβ FC of FM patients and the affective component of pain (SF-MPQ-2$_{affective}$ scaled to 10), with a trend to significance **(B)**. Within the mesiotemporal area, the basolateral amygdala (BLA) showed a significant FC decrease (blue color, $p<0.05$) **(C)**. Corresponding quantitative values were respectively (mean(SD)) -0.30(0.51) in patients (dark grey color and 0.11(0.44) in controls (light grey) **(D)**. In addition, FC was significantly correlated with SF-MPQ-2$_{affective}$ **(E)**. Grey dots correspond to patients with VAS$_d<3$ whereas black dots are related to patients with significant pain (VAS$_d\geq3$).

clinical basis for the affective dysfunction in FM can be demonstrated, involving a brain area eloquent for the expression and control of emotions, as well as the modulation of the affective dimension of pain [48]. Furthermore, the BLA receiving all sensory (including the nociceptive) inputs, is believed to add an emotional valence to the latter before conveying them onto the central nucleus of the amygdala, its main output region [49]. Differences and correlations to pain descriptors confined to the β oscillatory domain indicate a GABAergic dysfunction in FM [50]. Moreover, in accordance with previous hypotheses [9], these data further suggest Hβ modifications as an indicator of pain-related affective dysfunction involving the BLA (rich of GABAergic interneurons [51]) in FM.

The ability of surface EEG to probe amygdala activity is controversial, given the inherently low signal to noise ratio in deep brain structures. Recent data suggest that high density EEG can reliably sense subcortical electrophysiological activity (including in the amygdala) [52–57]. However, validation with intra-cortical recordings have only been obtained in studies using higher-density EEG montages [53,58], while we used only 64 electrodes. One should also be cautious extrapolating results from patients with coma [57] and epilepsy [56] to patients with fibromyalgia. Nevertheless, our data overall suggest a GABA-mediated functional disturbance of brain activity (pointing to the amygdala) related to the affective dimension of pain in FM. Alterations in brain function have previously been observed in FM studies assessing FC by other methods such as magnetoencephalography [59], or investigating different brain networks such as the default mode network or the salience network [3]. However, none of these studies associated observed electrophysiological changes to specific neurotransmitter pathways or to measures of the affective pain component. Nonetheless, they all add up to the evidence for existence of objective brain dysfunction in FM.

Although Lβ power correlated with ongoing pain intensity as previously observed [9], to our surprise, there was now an increase and a positive correlation, contrary to the previously

noticed decrease and negative correlation. Additionally, Lβ power maxima were previously observed in the posterior left-brain area (possibly corresponding to the somatosensory cortical areas) in contrast to the present Lβ power increase in the left PFC. Brain networks involved in pain chronification processes are similar to those implicated in executive functions (engaged in adaptive brain mechanisms) primarily controlled by the PFC [60], the latter being also implicated in pain regulation through abundant connections with the somatosensory system [61]. Considering the decrease in GABA-dependent inhibition occurring in chronic pain, we hypothesize that the increase in Lβ power possibly indicates a compensatory mechanism counteracting chronic pain-related GABAergic dysfunction. Interestingly, the PFC (in particular the dorsolateral PFC) constitutes a primary target for non-invasive brain therapies, such as the transcranial magnetic stimulation (TMS), including in FM [62]. Furthermore, TMS has been reported to have analgesic effects through GABAergic restauration [7]. Thus, the hypothesized "natural" GABA-related compensatory mechanisms would have potential analgesic effect, with possible reinforcement by therapeutic measures. Existing connections between the PFC and the amygdala [63] may finally provide an anatomic link between the identified site of brain dysfunction and the assumed compensatory region.

Previous investigations in FM reported both decrease [50] and increase [64] in GABAergic markers. In this study, Hβ decrease and Lβ increase were observed in different brain areas, associated to different pain descriptors, and differently interpreted (i.e. pathological decrease and compensatory increase), possibly reconciling these apparently conflicting results.

All depicted modifications in EEG markers occurred in left-sided brain areas in right-handed individuals, similar to previous observations in neuropathic pain and healthy populations [9], thereby supporting the concept of lateralization of chronic-pain-related brain modifications.

The specificity of the link between β EEG oscillations and pain can be questioned if we consider on one side, that GABAergic circuits also contribute to other oscillatory bands [7,65]. The current knowledge gives however, good indication for a link between fast EEG oscillations (namely β waves) and GABA concentrations in the brain [66,67]. Moreover, fast EEG oscillations are mainly driven by brain inhibitory interneurons, which are mostly GABAergic [67]. On the other side, we found an association between β waves and pain clinical descriptors, while at the same time, in the literature, β oscillations are linked with attention [68] or communication functions in verbal or non-verbal modalities [69]. Instead of seeing these findings as a contradiction with our results, we rather consider that β oscillations would indicate multifaceted aspects (including possibly the cognitive dimension) of pain. However, more investigations are warranted, to further enlighten this link. Additionally, due to the small number of participants in our research, further confirmation is necessary in larger studies. In addition, the interpretation frame of obtained results regarding the direction of electrophysiological modifications, their localization (i.e. decrease in the amygdala and increase in the PFC), as well as their implications in the mechanistic approach of pain in FM would need further investigation.

In conclusion, this study investigating EEG-measured GABAergic signaling modifications associated with the affective component of pain in FM patients, showed a Hβ FC decrease in the BLA correlated with the affective pain dimension, but an increase in PFC Lβ power associated with ongoing pain intensity. While the FC decrease was interpreted as part of the pathological process, the power increase was assumed to be compensatory, with potential therapeutic application. All disclosed modifications were left-sided, adding up to the emerging concept of left-lateralized changes in (GABAergic) pain-related brain pathways. Finally, given the small sample size and need for further methodological validation, larger and more accurate studies are needed to confirm these preliminary observations.

## Acknowledgments

We thank Prof Jean-Marie ANNONI for his helpful and meaningful comments at different steps of this study.

## Author Contributions

**Conceptualization:** Christian Mancini, Joelle Nsimire Chabwine.

**Data curation:** Soline Makowka, Lliure-Naima Mory, Michael Mouthon, Christian Mancini, Adrian G. Guggisberg, Joelle Nsimire Chabwine.

**Formal analysis:** Soline Makowka, Lliure-Naima Mory, Michael Mouthon, Adrian G. Guggisberg, Joelle Nsimire Chabwine.

**Investigation:** Lliure-Naima Mory, Christian Mancini, Joelle Nsimire Chabwine.

**Methodology:** Michael Mouthon, Christian Mancini, Joelle Nsimire Chabwine.

**Project administration:** Michael Mouthon, Christian Mancini, Joelle Nsimire Chabwine.

**Resources:** Michael Mouthon, Christian Mancini, Joelle Nsimire Chabwine.

**Software:** Adrian G. Guggisberg.

**Supervision:** Michael Mouthon, Christian Mancini, Joelle Nsimire Chabwine.

**Validation:** Michael Mouthon, Adrian G. Guggisberg, Joelle Nsimire Chabwine.

**Visualization:** Joelle Nsimire Chabwine.

**Writing – original draft:** Soline Makowka, Joelle Nsimire Chabwine.

**Writing – review & editing:** Soline Makowka, Lliure-Naima Mory, Michael Mouthon, Adrian G. Guggisberg, Joelle Nsimire Chabwine.

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
