## [Decision Letter · Decision Letter 0]

2 Feb 2022

PONE-D-21-31732Beta EEG left amygdala connectivity decreases and correlates with the affective pain in fibromyalgiaPLOS ONE

Dear Dr. Chabwine,

Thank you for submitting your manuscript to PLOS ONE. After careful consideration, we feel that it has merit but does not fully meet PLOS ONE’s publication criteria as it currently stands. Therefore, we invite you to submit a revised version of the manuscript that addresses the points raised during the review process.

Please include the following items when submitting your revised manuscript:A rebuttal letter that responds to each point raised by the academic editor and reviewer(s). You should upload this letter as a separate file labeled 'Response to Reviewers'.A marked-up copy of your manuscript that highlights changes made to the original version. You should upload this as a separate file labeled 'Revised Manuscript with Track Changes'.An unmarked version of your revised paper without tracked changes. You should upload this as a separate file labeled 'Manuscript'.

We look forward to receiving your revised manuscript.

Kind regards,

Claudia Sommer

Academic Editor

PLOS ONE

Journal Requirements:

Reviewers' comments:

Reviewer's Responses to Questions

**Comments to the Author**

1. Is the manuscript technically sound, and do the data support the conclusions?

Reviewer #1: Partly

Reviewer #2: No

2. Has the statistical analysis been performed appropriately and rigorously? 

Reviewer #1: Yes

Reviewer #2: No

3. Have the authors made all data underlying the findings in their manuscript fully available?

Reviewer #1: No

Reviewer #2: Yes

4. Is the manuscript presented in an intelligible fashion and written in standard English?

Reviewer #1: Yes

Reviewer #2: Yes

5. Review Comments to the Author

Reviewer #1: Title: Must be a pilot study in the title.

Study Design: Must define if it is correlational, or experimental study, etc.

Data collection: Must include the psychometric property of the tests and justify their use.

General data: Define gender: number of women?

Discussion: Describe more about the limitations of the study.

Reviewer #2: Using a 64-channel EEG, Makowka et al. investigated power and functional connectivity (FC) alterations in Fibromyalgia (FM) and related these to clinical parameters including sensory and affective pain components, anxiety, depression, and insomnia. In line with their hypotheses, the authors report decreased high beta connectivity in the basolateral amygdala in FM compared to healthy controls, the extent of which selectively correlated with questionnaire data assessing the affective pain component. Decreased beta band connectivity in the amygdala is thus interpreted as neural mechanism underlying the affective dysregulation seen in FM. In addition, the authors report an increase in relative low beta power in FM located in the left PFC which selectively correlated with pain intensity and might represent a compensatory mechanism.

Overall, the authors address a highly relevant research question using a timely analysis pipeline. By assessing FC patterns in FM, the presented study can contribute to current research efforts focusing on the development of brain-based biomarkers of chronic pain which remains a key challenge at the intersection of cognitive and clinical science. However, major concerns regarding the methodological rigor and especially the sample size of the study should be addressed before a final evaluation can be made.

Major Comments

1. It is questionable whether the sample size of N= 16 patients is sufficient to detect the correlation of interest in a reliable and replicable fashion. Despite the absence of a correction for multiple comparisons several p-values are almost non-significant (e.g., p = 0.049 for the correlation between amygdala connectivity and the affective pain component) and I am very concerned that these findings might represent false positives that would not replicate in another data set. To enhance the confidence in their findings, I strongly recommend that the authors conduct sample size calculations and adjust the sample size accordingly. In addition, the authors should consider correcting for multiple comparisons, e.g., by taking the 5 frequency bands investigated into account.

2. As mentioned in the discussion, EEG is limited in its spatial resolution, and it is highly debatable whether it can provide accurate information regarding deep and focal sources such as the amygdala. The authors cite studies exploring this question, however, the cited papers used 128 and 256 electrodes, respectively and are therefore not directly comparable with the current 64-electrode montage. Consequently, this limitation should be discussed more prominently and maybe also warrants a more cautious choice of title and wording in the abstract, results, and discussion section.

3. Information regarding the medication of included patients is missing and should be added to the manuscript.

4. Several aspects of the EEG data analysis section require further clarification:

• How many data segments were rejected/retained for analysis?

• The investigated beta sub-bands deviate from the conventional canonical frequency bands (e.g., Pernet et al., Nat. Neurosci, 2020). Thus, it would be important to address the question whether findings replicate when performing a re-analysis using the conventional frequency boundaries.

• Is it correct that power analyses were also conducted in source space? If so, this should be stated more clearly.

• Which toolboxes/software were used for the calculation of power, FC, and the weighted node degree?

• Which toolboxes/software packages were used for statistical analyses?

• The FC measure calculated was probably the imaginary component of the complex valued coherency and not coherence (e.g., see Bastos, Front. Syst. Neurosci, 2016)? If so, this should be adjusted throughout the manuscript.

• The FC region of interest analysis should be described in more detail. How were regions of interest (such as the BLA) motivated? Was this done a priori or based on the results of the cluster-based permutation tests?

Minor Comments

1. Were the hypotheses mentioned in the introduction and the analysis plan preregistered?

2. How was the recording duration motivated (20 min seems rather long and might lead to sleep artifacts depending on the seating position) and why was it approximately 20 min implying variations between participants?

3. Previous studies examining FC changes in FM (e.g., Hsiao et al., J. Headache Pain, 2017; Vanneste et al., PLoS One, 2017) could be included in the discussion.

6. PLOS authors have the option to publish the peer review history of their article (what does this mean?). If published, this will include your full peer review and any attached files.

Reviewer #1: No

Reviewer #2: No

---

## [Author Response · Author response to Decision Letter 0]

3 Oct 2022

Reviewer #1:

A- Title: Must be a pilot study in the title.

Following the reviewer’s remark and in accordance with the journal’s guidelines, the manuscript’s title has now been changed into: “EEG beta functional connectivity decrease in the left amygdala correlates with affective pain in fibromyalgia: a pilot study” (lines 1-3).

B- Study Design: Must define if it is correlational, or experimental study, etc.

We thank the reviewer for raising our awareness on this important point. In fact, the primary goal of this study was to find an electrophysiological correlate of the affective pain component in the beta band. Thus, this is a correlational cross-sectional case-control study. 

We have now better specified the aim and the design of the study respectively in the abstract, in the introduction and in the methods section (lines 27-28, 72, 75). 

C- Data collection: Must include the psychometric property of the tests and justify their use.

All tests (VAS, SF-MPQ-2, HADS and ISI) are validated and widely used in chronic pain patients (including those suffering from fibromyalgia). Thus, in general, their psychometric properties are no longer detailed in publications. Nevertheless, we provide below some illustrative literature and have included in the manuscript, as requested by the reviewer, a summarized information regarding the validity of those tests and the most important references supporting their use in chronic pain/fibromyalgia patients (lines 111-122). 

McGill Pain Questionnaire the Short-Form-2 (MPQ-SF-2)

- Reliable and valid in diverse chronic pain syndromes with large samples (1, 2); 

- Use in fibromyalgia patients (3, 4).

Visual Analog Scale (VAS)

- Good statistical and psychometric properties in chronic pain patients (5-8); 

- Use in chronic pain patients, including those suffering from fibromyalgia (9).

Hospital Anxiety and Depression Scale (HADS)

- Valid psychometric properties (10, 11) ;

- Use in fibromyalgia patients (12-14).

Insomnia Severity Index (ISI)

- Reliable and valid in clinical settings (15, 16);

- Use in clinical pain, including in fibromyalgia (17, 18). 

Combination of scales:

- SF_MPQ-2 and HADS in fibromyalgia (19); 

- SF-MPQ-2 and VAS in fibromyalgia (3, 20) ;

- ISI, HADS, SF-MPQ in chronic pain patients (21).

D- General data: Define gender: number of women?

According to our protocol, the study included only patients of female sex. The reviewer is right in that the use of the term “gender” instead of “sex” could be confusing. Therefore, we now only use the word “sex” in the manuscript (line 30-31, 83, 101).

E- Discussion: Describe more about the limitations of the study.

Study limitations are detailed further in the manuscript in the discussion paragraph (line 283-287). The results of this pilot study need further confirmation in larger settings. In addition, the interpretation frame and raised hypotheses should be further investigated and confirmed. Specifically, the direction of EEG modifications and their localization, as well as their implications in the mechanistic and therapeutic approach of pain in fibromyalgia could be further investigated. 

Reviewer #2: 

Major Comments

1.a - It is questionable whether the sample size of N= 16 patients is sufficient to detect the correlation of interest in a reliable and replicable fashion. 

To enhance the confidence in their findings, I strongly recommend that the authors conduct sample size calculations and adjust the sample size accordingly. 

The main goal of this pilot study was to find an electrophysiological correlate of the affective pain component in fibromyalgia. Previous studies have reported that functional connectivity correlates with clinical impairments such as motor and cognitive deficits, with correlation coefficients around 0.6 – 0.7 (22-24). Thus, based on these findings, we expected similar correlation coefficients in the present study and performed a power analysis revealing that a sample size of 16 patients would give 80% power to significantly detect comparable correlation (correlation coefficient of 0.65 at p<0.05) (25). An indication on the power calculation was added in the methods section (line 89-93) as well as in the discussion (line 282-283).

1.b In addition, the authors should consider correcting for multiple comparisons, e.g., by taking the 5 frequency bands investigated into account.

This study did not equally and simultaneously explore all the 5 EEG frequency bands. Rather, there was an a priori hypothesis that the expected differences would occur in the beta band, based on the suspected GABAergic mechanisms. There is good evidence in the literature, that brain GABAergic neurotransmission driven by cortical interneurons, is mainly represented in the beta-band (26-30). The delta-band was used as a control band to demonstrate that our findings were specific to the beta band. The remaining frequencies were merely used for computing the relative power of the beta band, i.e., the power of the beta band with regards to the distribution of all bands. Thus, in this respect, there was no need to correct for multiple testing.

We acknowledge however, that we did not control for testing the two sub-bands in the beta frequency range when comparing patients to controls. Nevertheless, the correlation found between one of the two sub-bands (high beta) and the clinical variable of interest (affective pain), in addition to the decrease observed in the same band within an eloquent brain area for that clinical variable (amygdala), suggest that these findings are meaningful. Furthermore, in order to estimate the probability of finding such a difference between groups in any of two frequency bands plus a correlation with a clinical variable by pure chance, we performed a small simulation with random numbers. Out of 2000 iterations of randomly generated numbers (randn function in MATLAB) of the same size as in our real dataset, we obtained a significant difference between groups at any of 2 random vectors AND a correlation with a third random vector in only 8 cases. Thus, the probability that our findings as a whole are due to mere chance can be estimated to about 8/2000, i.e., p=0.004. 

1.c - Despite the absence of a correction for multiple comparisons several p-values are almost non-significant (e.g., p = 0.049 for the correlation between amygdala connectivity and the affective pain component) and I am very concerned that these findings might represent false positives that would not replicate in another data set.

As written in the analysis paragraph of the manuscript, correction for multiple testing was performed (line 163). For more clarity, we have now specified the method used for the multiple testing (cluster-size threshold, line 163-165).

The reviewer is right in saying that the correlation p value is close to the limit of significance (<.05). However, the congruence of our findings described above, in addition to the simulation excluding random observations, suggests that this is an observation worth paying attention, opening a new avenue of research in fibromyalgia for further exploration. 

2. As mentioned in the discussion, EEG is limited in its spatial resolution, and it is highly debatable whether it can provide accurate information regarding deep and focal sources such as the amygdala. 

The authors cite studies exploring this question, however, the cited papers used 128 and 256 electrodes, respectively and are therefore not directly comparable with the current 64-electrode montage. Consequently, this limitation should be discussed more prominently and maybe also warrants a more cautious choice of title and wording in the abstract, results, and discussion section.

We agree that the use of a 64-electrode montage is not the highest solution for spatial resolution, but constitutes, as published in reference textbooks, the minimal adequate layout for a reliable source reconstruction (31). Furthermore, EEG recordings with less than 64 channels could target the amygdala in clinical populations (32, 33). Finally, from a theoretical perspective, we expect that a reduction of the number of electrodes reduces the spatial sampling, but not per se the ability to capture deep sources. Thus, the source needs to be larger, but not necessarily less deep (34).

Overall, we assume from above-mentioned publications, that this spatial coverage was enough to respond to the question addressed by our manuscript. We have now also added references with similar spatial resolution in the discussion section (line 238). 

3. Information regarding the medication of included patients is missing and should be added to the manuscript.

The patients’ medication was added to the manuscript in the form of a table (Table 1, line 185).

4. Several aspects of the EEG data analysis section require further clarification:

4.a How many data segments were rejected/retained for analysis?

The whole recording time lasted 24 minutes, but we retained the first 5 minutes of artifact-free data. We added this information on the method section (line 128-130).

4.b The investigated beta sub-bands deviate from the conventional canonical frequency bands (e.g., Pernet et al., Nat. Neurosci, 2020). Thus, it would be important to address the question whether findings replicate when performing a re-analysis using the conventional frequency boundaries.

The frequency range of interest was the beta band, that we defined between 13 and 30 Hz, as recommended by Pernet et al. (2020) (35). The investigated beta sub-bands (low beta, 13-20 Hz, and high beta, 20-30 Hz), were chosen according to our previous findings (26), as mentioned in the method section (line 146-148). Moreover, those frequency boundaries are often used in EEG research (36, 37). 

4.c - Is it correct that power analyses were also conducted in source space? If so, this should be stated more clearly.

This information was mentioned in the result section, but indeed lacked in the methods section. We have now stated it also in the methods section (line 149). 

4.d - Which toolboxes/software were used for the calculation of power, FC, and the weighted node degree?

For the EEG data analysis (FC, weighted node degree), this is mentioned in the Methods section of the paper: “Analyses were performed in MATLAB (The MathWorks), using the toolbox NUTMEG” (line 138-139). 

The calculation of power was performed using the online Sample Size Calculators from Kohn and Senyak (2021) (25). This reference was added on the method section (line 93). 

4.e Which toolboxes/software packages were used for statistical analyses?

Voxel-wise statistics were performed with NUTMEG, the remaining analyses with the Statistics toolbox of Matlab. This information was added on the method section (line 171-172).

4.f The FC measure calculated was probably the imaginary component of the complex valued coherency and not coherence (e.g., see Bastos, Front. Syst. Neurosci, 2016)? If so, this should be adjusted throughout the manuscript.

According to the nomenclature of the original description of the imaginary component of coherence (38), the term “coherence” refers to absolute values, “coherency” to the raw values that range between -1 and 1. As we used the absolute values, we prefer to use the term “coherence”, in accordance with the original terminology (line 154-156).

4.g The FC region of interest analysis should be described in more detail. How were regions of interest (such as the BLA) motivated? Was this done a priori or based on the results of the cluster-based permutation tests?

We did not predefine any region of interest in the analysis presented in the paper. The FC difference in the mesiotemporal lobe came out of the cluster-based permutation test. Based on an extensive literature on the importance of the BLA in affective pain (39-41), we then investigated whether this subregion within the mesiotemporal lobe also revealed a beta-band FC correlate of fibromyalgia, in particular the affective pain component. Indeed, the frequency band for which differences were observed witnessed correlation with our clinical variable of interest, highly suggesting that our results could be clinically meaningful.

Minor Comments

1. Were the hypotheses mentioned in the introduction and the analysis plan preregistered?

No, this study was not preregistered, because it is part of a precedent larger project investigating GABAergic markers of chronic pain, which was not preregistered. 

2. How was the recording duration motivated (20 min seems rather long and might lead to sleep artifacts depending on the seating position) and why was it approximately 20 min implying variations between participants?

The recording time was exactly of 24 minutes for each participant, thus there was no variation between participants. We corrected the manuscript accordingly (line 128). The recording length was motivated by the will to obtain enough “clean” data (epochs) for the analysis. We agree with the reviewer that this duration is long and keen to sleepiness. For this reason, the participants were maintained seated, we used acoustic sounds two times during the recordings and finally selected only the 5 first minutes of clean recording (line 128-130).

3. Previous studies examining FC changes in FM (e.g., Hsiao et al., J. Headache Pain, 2017; Vanneste et al., PLoS One, 2017) could be included in the discussion.

The two mentioned studies were added and referenced to in the discussion, as additional evidence for objective modifications of brain function in the context of fibromyalgia (42, 43) (line 240-246). 

We hope our manuscript will be now found suitable for publication in PLOS One.

Yours sincerely,

Joelle N. CHABWINE, MD, PhD

REFERENCES

1. Dworkin RH, Turk DC, Revicki DA, Harding G, Coyne KS, Peirce-Sandner S, et al. Development and initial validation of an expanded and revised version of the Short-form McGill Pain Questionnaire (SF-MPQ-2). Pain. 2009;144(1-2):35-42.

2. Dworkin RH, Turk DC, Trudeau JJ, Benson C, Biondi DM, Katz NP, et al. Validation of the Short-form McGill Pain Questionnaire-2 (SF-MPQ-2) in acute low back pain. J Pain. 2015;16(4):357-66.

3. Geisser ME, Gracely RH, Giesecke T, Petzke FW, Williams DA, Clauw DJ. The association between experimental and clinical pain measures among persons with fibromyalgia and chronic fatigue syndrome. Eur J Pain. 2007;11(2):202-7.

4. Harris RE, Gracely RH, McLean SA, Williams DA, Giesecke T, Petzke F, et al. Comparison of clinical and evoked pain measures in fibromyalgia. J Pain. 2006;7(7):521-7.

5. Price DD, McGrath PA, Rafii A, Buckingham B. The validation of visual analogue scales as ratio scale measures for chronic and experimental pain. Pain. 1983;17(1):45-56.

6. Hawker GA, Mian S, Kendzerska T, French M. Measures of adult pain: Visual Analog Scale for Pain (VAS Pain), Numeric Rating Scale for Pain (NRS Pain), McGill Pain Questionnaire (MPQ), Short-Form McGill Pain Questionnaire (SF-MPQ), Chronic Pain Grade Scale (CPGS), Short Form-36 Bodily Pain Scale (SF-36 BPS), and Measure of Intermittent and Constant Osteoarthritis Pain (ICOAP). Arthritis Care Res (Hoboken). 2011;63 Suppl 11:S240-52.

7. Hjermstad MJ, Fayers PM, Haugen DF, Caraceni A, Hanks GW, Loge JH, et al. Studies comparing Numerical Rating Scales, Verbal Rating Scales, and Visual Analogue Scales for assessment of pain intensity in adults: a systematic literature review. J Pain Symptom Manage. 2011;41(6):1073-93.

8. Ferraz MB, Quaresma MR, Aquino LR, Atra E, Tugwell P, Goldsmith CH. Reliability of pain scales in the assessment of literate and illiterate patients with rheumatoid arthritis. J Rheumatol. 1990;17(8):1022-4.

9. Cheatham SW, Kolber MJ, Mokha M, Hanney WJ. Concurrent validity of pain scales in individuals with myofascial pain and fibromyalgia. J Bodyw Mov Ther. 2018;22(2):355-60.

10. Bjelland I, Dahl AA, Haug TT, Neckelmann D. The validity of the Hospital Anxiety and Depression Scale. An updated literature review. J Psychosom Res. 2002;52(2):69-77.

11. Zigmond AS, Snaith RP. The hospital anxiety and depression scale. Acta Psychiatr Scand. 1983;67(6):361-70.

12. Nam S, Tin D, Bain L, Thorne JC, Ginsburg L. Clinical utility of the Hospital Anxiety and Depression Scale (HADS) for an outpatient fibromyalgia education program. Clin Rheumatol. 2014;33(5):685-92.

13. Vallejo MA, Rivera J, Esteve-Vives J, Rodriguez-Munoz MF, Grupo I. [Use of the Hospital Anxiety and Depression Scale (HADS) to evaluate anxiety and depression in fibromyalgia patients]. Rev Psiquiatr Salud Ment. 2012;5(2):107-14.

14. Marchi L, Marzetti F, Orru G, Lemmetti S, Miccoli M, Ciacchini R, et al. Alexithymia and Psychological Distress in Patients With Fibromyalgia and Rheumatic Disease. Front Psychol. 2019;10:1735.

15. Morin CM, Belleville G, Belanger L, Ivers H. The Insomnia Severity Index: psychometric indicators to detect insomnia cases and evaluate treatment response. Sleep. 2011;34(5):601-8.

16. Bastien CH, Vallieres A, Morin CM. Validation of the Insomnia Severity Index as an outcome measure for insomnia research. Sleep Med. 2001;2(4):297-307.

17. Aloush V, Gurfinkel A, Shachar N, Ablin JN, Elkana O. Physical and mental impact of COVID-19 outbreak on fibromyalgia patients. Clin Exp Rheumatol. 2021;39 Suppl 130(3):108-14.

18. Gammoh OS, Al-Smadi A, Tayfur M, Al-Omari M, Al-Katib W, Zein S, et al. Syrian female war refugees: preliminary fibromyalgia and insomnia screening and treatment trends. Int J Psychiatry Clin Pract. 2020;24(4):387-91.

19. Sanchez AI, Martinez MP, Miro E, Medina A. Predictors of the pain perception and self-efficacy for pain control in patients with fibromyalgia. Span J Psychol. 2011;14(1):366-73.

20. da Cunha Ribeiro RP, Franco TC, Pinto AJ, Pontes Filho MAG, Domiciano DS, de Sa Pinto AL, et al. Prescribed Versus Preferred Intensity Resistance Exercise in Fibromyalgia Pain. Front Physiol. 2018;9:1097.

21. Tang NK, Wright KJ, Salkovskis PM. Prevalence and correlates of clinical insomnia co-occurring with chronic back pain. J Sleep Res. 2007;16(1):85-95.

22. Dubovik S, Pignat JM, Ptak R, Aboulafia T, Allet L, Gillabert N, et al. The behavioral significance of coherent resting-state oscillations after stroke. Neuroimage. 2012;61(1):249-57.

23. Allaman L, Mottaz A, Kleinschmidt A, Guggisberg AG. Spontaneous Network Coupling Enables Efficient Task Performance without Local Task-Induced Activations. J Neurosci. 2020;40(50):9663-75.

24. Guggisberg AG, Rizk S, Ptak R, Di Pietro M, Saj A, Lazeyras F, et al. Two intrinsic coupling types for resting-state integration in the human brain. Brain Topogr. 2015;28(2):318-29.

25. Kohn M, Senyak J. Sample Size Calculators. Website. 2021.

26. Teixeira M, Mancini C, Wicht CA, Maestretti G, Kuntzer T, Cazzoli D, et al. Beta Electroencephalographic Oscillation Is a Potential GABAergic Biomarker of Chronic Peripheral Neuropathic Pain. Front Neurosci. 2021;15:594536.

27. Baumgarten TJ, Oeltzschner G, Hoogenboom N, Wittsack HJ, Schnitzler A, Lange J. Beta Peak Frequencies at Rest Correlate with Endogenous GABA+/Cr Concentrations in Sensorimotor Cortex Areas. PLoS One. 2016;11(6):e0156829.

28. Christian EP, Snyder DH, Song W, Gurley DA, Smolka J, Maier DL, et al. EEG-beta/gamma spectral power elevation in rat: a translatable biomarker elicited by GABA(Aalpha2/3)-positive allosteric modulators at nonsedating anxiolytic doses. J Neurophysiol. 2015;113(1):116-31.

29. Barr MS, Farzan F, Davis KD, Fitzgerald PB, Daskalakis ZJ. Measuring GABAergic inhibitory activity with TMS-EEG and its potential clinical application for chronic pain. J Neuroimmune Pharmacol. 2013;8(3):535-46.

30. Jones EG. GABAergic neurons and their role in cortical plasticity in primates. Cereb Cortex. 1993;3(5):361-72.

31. Michel C, Koenig T, Brandeis D, Gianotti L, Wackermann J. Electrical Neuroimaging. Cambridge: Cambridge University Press. 2009:page 83.

32. De Stefano P, Carboni M, Pugin D, Seeck M, Vulliemoz S. Brain networks involved in generalized periodic discharges (GPD) in post-anoxic-ischemic encephalopathy. Resuscitation. 2020;155:143-51.

33. Sharma P, Scherg M, Pinborg LH, Fabricius M, Rubboli G, Pedersen B, et al. Ictal and interictal electric source imaging in pre-surgical evaluation: a prospective study. Eur J Neurol. 2018;25(9):1154-60.

34. Michel CM, Brunet D. EEG Source Imaging: A Practical Review of the Analysis Steps. Front Neurol. 2019;10:325.

35. Pernet C, Garrido MI, Gramfort A, Maurits N, Michel CM, Pang E, et al. Issues and recommendations from the OHBM COBIDAS MEEG committee for reproducible EEG and MEG research. Nat Neurosci. 2020;23(12):1473-83.

36. von Rotz R, Kometer M, Dornbierer D, Gertsch J, Salome Gachet M, Vollenweider FX, et al. Neuronal oscillations and synchronicity associated with gamma-hydroxybutyrate during resting-state in healthy male volunteers. Psychopharmacology (Berl). 2017;234(13):1957-68.

37. Haenschel C, Baldeweg T, Croft RJ, Whittington M, Gruzelier J. Gamma and beta frequency oscillations in response to novel auditory stimuli: A comparison of human electroencephalogram (EEG) data with in vitro models. Proc Natl Acad Sci U S A. 2000;97(13):7645-50.

38. Nolte G, Bai O, Wheaton L, Mari Z, Vorbach S, Hallett M. Identifying true brain interaction from EEG data using the imaginary part of coherency. Clin Neurophysiol. 2004;115(10):2292-307.

39. Corder G, Ahanonu B, Grewe BF, Wang D, Schnitzer MJ, Scherrer G. An amygdalar neural ensemble that encodes the unpleasantness of pain. Science. 2019;363(6424):276-81.

40. Seno MDJ, Assis DV, Gouveia F, Antunes GF, Kuroki M, Oliveira CC, et al. The critical role of amygdala subnuclei in nociceptive and depressive-like behaviors in peripheral neuropathy. Sci Rep. 2018;8(1):13608.

41. Thompson JM, Neugebauer V. Amygdala Plasticity and Pain. Pain Res Manag. 2017;2017:8296501.

42. Hsiao FJ, Wang SJ, Lin YY, Fuh JL, Ko YC, Wang PN, et al. Altered insula-default mode network connectivity in fibromyalgia: a resting-state magnetoencephalographic study. J Headache Pain. 2017;18(1):89.

43. Vanneste S, Ost J, Van Havenbergh T, De Ridder D. Resting state electrical brain activity and connectivity in fibromyalgia. PLoS One. 2017;12(6):e0178516.

---

## [Decision Letter · Decision Letter 1]

16 Nov 2022

PONE-D-21-31732R1EEG Beta functional connectivity decrease in the left amygdala correlates with the affective pain in fibromyalgia : a pilot studyPLOS ONE

Dear Dr. Chabwine,

Thank you for submitting your manuscript to PLOS ONE. After careful consideration, we feel that it has merit but does not fully meet PLOS ONE’s publication criteria as it currently stands. Therefore, we invite you to submit a revised version of the manuscript that addresses the points raised during the review process. Please be aware that this should be the final round of revision.

We look forward to receiving your revised manuscript.

Kind regards,

Claudia Sommer

Academic Editor

PLOS ONE

Reviewers' comments:

Reviewer's Responses to Questions

**Comments to the Author**

1. If the authors have adequately addressed your comments raised in a previous round of review and you feel that this manuscript is now acceptable for publication, you may indicate that here to bypass the “Comments to the Author” section, enter your conflict of interest statement in the “Confidential to Editor” section, and submit your "Accept" recommendation.

Reviewer #1: All comments have been addressed

Reviewer #2: (No Response)

2. Is the manuscript technically sound, and do the data support the conclusions?

Reviewer #1: Partly

Reviewer #2: Partly

3. Has the statistical analysis been performed appropriately and rigorously? 

Reviewer #1: Yes

Reviewer #2: Yes

4. Have the authors made all data underlying the findings in their manuscript fully available?

Reviewer #1: No

Reviewer #2: No

5. Is the manuscript presented in an intelligible fashion and written in standard English?

Reviewer #1: Yes

Reviewer #2: Yes

6. Review Comments to the Author

Reviewer #1: Considering that this is a pilot study, it can be accepted.

However, the larger study must overcome the limitations pointed out by the second reviewer.

Reviewer #2: Evaluation

The authors have addressed several of my concerns and added relevant information to the methods section. However, my two main concerns remain, namely (1) that the sample size might be too low for robust results and (2) that 64-channel EEG recordings may not provide the resolution needed to accurately detect functional connectivity changes in the amygdala. Thus, it is crucial to transparently communicate these limitations in the manuscript.

Sample size:

The sample size limitation is now prominently communicated by describing the study as a pilot study in the title. In addition, the authors report the a posteriori power calculation I mentioned in the first review. However, the presented power calculation is problematic because the underlying effect size estimate of r = 0.65 was derived from previous studies examining very different (patient) populations (patients with a stroke in two studies and healthy participants in one study), and thus, might likely not be appropriate. In absence of a better estimate, I recommend deleting the corresponding paragraph from the manuscript.

Spatial resolution:

Indeed, EEG source reconstruction is possible with 64 electrodes, but the relevant question in this context is whether 64 electrodes suffice to reliably detect connectivity changes in the amygdala which is both a deep and a small structure. The authors list two studies examining amygdala activity in clinical populations with less than 64 EEG channels. However, these studies examined different populations (comatose patients and patients with epilepsy) and did not validate their approach (e.g., through simultaneous intracortical recordings). Such validation approaches have yielded promising evidence for high-density EEG (256 electrodes) and MEG (see Lopes da Silva, Brain Topogr, 2019 for a commentary), but I am not aware of similar publications for 64-channel EEG. Thus, in absence of convincing validation studies, it is important that this limitation is not downplayed (page 11, line 240), but prominently discussed, e.g., on page 11 and 13.

Minor comments:

• Page 6, line 132: specify that the first five minutes of artifact-free data were retained

• Page 11, line 239: delete “s” (“recent data”)

• Page 13, line 285: further confirmation “is” necessary instead of “would be”

7. PLOS authors have the option to publish the peer review history of their article (what does this mean?). If published, this will include your full peer review and any attached files.

Reviewer #1: **Yes: **Susana Cardoso

Reviewer #2: No

---

## [Author Response · Author response to Decision Letter 1]

23 Dec 2022

Answers to the Reviewers, second round of revisions

Dear Prof Sommer,

We thank you for the opportunity to revise a second time our manuscript and improve it according to the reviewers’ new remarks. We also thank the reviewers for carefully reading all our answers and modifications made to the manuscript, as well as for their comments and advices that are clearly and positively oriented to enrich the manuscript and make it more accurate. We addressed their remarks at our best. In addition, we took this opportunity to review the whole manuscript in search of remaining typos and other minor errors (these modifications are highlighted in orange). In this respect, we removed the reference 11, which was not any more cited (page 16, line 344), while overlooked double citations were removed (page 20, line 449 and line 466; page 21 line 469 and line 482; page 23 line 526).

Reviewer #1 acknowledged that our answers addressed all his queries. Our answers to reviewer #2 are listed below. Corrections in the manuscript are highlighted in blue. Page and line references mentioned below are always related to “track-change” versions of the manuscript. 

Reviewer #2: Evaluation

The authors have addressed several of my concerns and added relevant information to the methods section. However, my two main concerns remain, namely (1) that the sample size might be too low for robust results and (2) that 64-channel EEG recordings may not provide the resolution needed to accurately detect functional connectivity changes in the amygdala. Thus, it is crucial to transparently communicate these limitations in the manuscript.

1- Sample size:

The sample size limitation is now prominently communicated by describing the study as a pilot study in the title. In addition, the authors report the a posteriori power calculation I mentioned in the first review. However, the presented power calculation is problematic because the underlying effect size estimate of r = 0.65 was derived from previous studies examining very different (patient) populations (patients with a stroke in two studies and healthy participants in one study), and thus, might likely not be appropriate. In absence of a better estimate, I recommend deleting the corresponding paragraph from the manuscript.

We removed the paragraph related to the power computation (page 4 line 88-and changed the sentence related to it in the discussion: 

Page 13, line 287-288: Additionally, due to the small number of participants in our research, further confirmation is necessary in larger studies.

2- Spatial resolution:

Indeed, EEG source reconstruction is possible with 64 electrodes, but the relevant question in this context is whether 64 electrodes suffice to reliably detect connectivity changes in the amygdala which is both a deep and a small structure. The authors list two studies examining amygdala activity in clinical populations with less than 64 EEG channels. However, these studies examined different populations (comatose patients and patients with epilepsy) and did not validate their approach (e.g., through simultaneous intracortical recordings). Such validation approaches have yielded promising evidence for high-density EEG (256 electrodes) and MEG (see Lopes da Silva, Brain Topogr, 2019 for a commentary), but I am not aware of similar publications for 64-channel EEG. Thus, in absence of convincing validation studies, it is important that this limitation is not downplayed (page 11, line 240), but prominently discussed, e.g., on page 11 and 13.

We agree with the reviewer that both cited EEG studies evaluating electrical activity in the amygdala with less than 64 electrodes have not been validated with intracortical recordings. Therefore, we have toned down our interpretation of our findings, as follows: 

Page 11, line 236-242: The ability of surface EEG to probe amygdala activity is controversial, given the inherently low signal to noise ratio in deep brain structures. Recent data suggest that high density EEG can reliably sense subcortical electrophysiological activity (including in the amygdala) [53-58]. However, validation with intra-cortical recordings have only been obtained in studies using higher-density EEG montages [54,59], while we used only 64 electrodes. One should also be cautious extrapolating results from patients with coma [58] and epilepsy [57] to patients with fibromyalgia.

Page 14, line 299-301: Finally, given the small sample size and need for further methodological validation, larger and more accurate studies are needed to confirm these preliminary observations.

4 - Page 6, line 132: specify that the first five minutes of artifact-free data were retained

The mention of this information was initially put in the wrong section (page 6, line 127-128). Now, we have deleted it on this location and displaced it to the correct place (page 6, line 134-135). 

5- Page 11, line 239: delete “s” (“recent data”)

Corrected (page 11, line 237)

6- Page 13, line 285: further confirmation “is” necessary instead of “would be”

Corrected (page 13, line 303-304)

We hope we have satisfactorily addressed the reviewer’s concerns and that our manuscript will be now found suitable for publication in PLOS One.

Yours sincerely,

Joelle N. CHABWINE, MD, PhD

---

## [Decision Letter · Decision Letter 2]

7 Feb 2023

EEG Beta functional connectivity decrease in the left amygdala correlates with the affective pain in fibromyalgia : a pilot study

PONE-D-21-31732R2

Dear Dr. Chabwine,

We’re pleased to inform you that your manuscript has been judged scientifically suitable for publication and will be formally accepted for publication once it meets all outstanding technical requirements.

Kind regards,

Claudia Sommer

Academic Editor

PLOS ONE

Additional Editor Comments (optional):

Reviewers' comments:

Reviewer's Responses to Questions

**Comments to the Author**

1. If the authors have adequately addressed your comments raised in a previous round of review and you feel that this manuscript is now acceptable for publication, you may indicate that here to bypass the “Comments to the Author” section, enter your conflict of interest statement in the “Confidential to Editor” section, and submit your "Accept" recommendation.

Reviewer #2: All comments have been addressed

2. Is the manuscript technically sound, and do the data support the conclusions?

Reviewer #2: Yes

3. Has the statistical analysis been performed appropriately and rigorously? 

Reviewer #2: Yes

4. Have the authors made all data underlying the findings in their manuscript fully available?

Reviewer #2: No

5. Is the manuscript presented in an intelligible fashion and written in standard English?

Reviewer #2: Yes

6. Review Comments to the Author

Reviewer #2: (No Response)

7. PLOS authors have the option to publish the peer review history of their article (what does this mean?). If published, this will include your full peer review and any attached files.

Reviewer #2: No

---

## [Editor Report · Acceptance letter]

10 Feb 2023

PONE-D-21-31732R2 

EEG Beta functional connectivity decrease in the left amygdala correlates with the affective pain in fibromyalgia: a pilot study 

Dear Dr. Chabwine:

I'm pleased to inform you that your manuscript has been deemed suitable for publication in PLOS ONE. Congratulations! Your manuscript is now with our production department. 

Kind regards, 

on behalf of

Prof. Dr. Claudia Sommer 

Academic Editor

PLOS ONE